# Light Yield Response of Neutron Scintillation Screens to Sudden Flux Changes

**DOI:** 10.3390/jimaging6120134

**Published:** 2020-12-05

**Authors:** Tobias Neuwirth, Bernhard Walfort, Simon Sebold, Michael Schulz

**Affiliations:** 1Heinz Maier-Leibnitz Zentrum (MLZ), Lichtenbergstr. 1, D-85748 Garching, Germany; Simon.Sebold@frm2.tum.de (S.S.); Michael.Schulz@frm2.tum.de (M.S.); 2RC Tritec AG, Speicherstrasse 60 a, CH-9053 Teufen, Switzerland; walfort@rctritec.com

**Keywords:** high frame rate neutron imaging, scintillation screens, decay time, neutron imaging

## Abstract

We performed a study of the initial and long term light yield of different scintillation screen mixtures for neutron imaging during constant neutron irradiation. We evaluated the light yield during different neutron flux levels as well as at different temperatures. As high frame rate imaging is a topic of interest in the neutron imaging community, the decay characteristics of scintillation screens are of interest as well. Hence, we also present and discuss the decay behavior of the different scintillation screen mixtures on a time scale of seconds. We have found that the decay time of ZnS:Cu/^6^LiF excited with a high neutron flux is potentially much longer than typically stated. While most of the tested scintillation screens do not provide a significant improvement over currently used scintillation screen materials, Zn(Cd)S:Ag/^6^LiF seems to be a good candidate for high frame rate imaging due to its high light yield, long-term stability as well as fast decay compared to the other evaluated scintillation screens.

## 1. Introduction

Nondestructive testing is a key feature of neutron imaging [1]. The neutron contrast modality depends on the specific nuclear core structure, enabling the analysis of both heavy and light elements, as well as differentiation between isotopes. Hence, neutron imaging has been established as an efficient technique in materials science, archaeology and engineering, whenever X-rays do not provide enough contrast. In recent times, one focus of neutron imaging is time-resolved imaging in the sub-second regime [2,3,4]. Exposure times in this regime pose challenges with respect to the available neutron flux, the camera system, as well as the decay time of the scintillation materials, which are an essential component of almost all detection systems for neutron imaging. A shorter exposure time causes, at a particular neutron flux, less photons to be detected in one image, leading to increased Poisson noise in the images. Furthermore, the ratio between the generated signal and the thermal noise in the camera has to be high enough to be able to distinguish signal from noise. The decay time of the light emission of scintillation screens influences the following images, as the light emission generated in one frame may still be emitted during the exposure of the subsequent frame i.e., generating ghost images. This degrades the quality of the measured data. Rapidly decaying scintillation screens are needed to suppress this effect. In our experiments, one of the standard scintillation screen mixtures (ZnS:Cu/^6^LiF) shows a relatively slow decay time of approximately 2.5±0.5
s to 1% of the light yield after the end of irradiation with neutrons. It should be noted that this result differs from previously recorded results, which suggest a decay time of 85 μs to 10% [5] in a ZnS/^6^LiF scintillation screen. We expect this deviation to be a result of the different excitation radiations used in the experiments. In the discussion, we give a more detailed explanation of the origin of this discrepancy.

While various studies of the decay of neutron scintillation screens have been performed in the past [6,7], those studies investigated the decay in the μs-range. In recent times, the focus of scintillation screen development has been on high spatial resolution, where multiple studies have been performed [8,9,10]. New scintillation screens specifically for γ- discrimination have also been studied [11,12]. In these studies the decay time of the scintillation screens has been examined as well. In this publication, we characterize the time and flux dependent light yield characteristics of five new commercially available scintillation screen mixtures on a time scale of seconds and compare them with standard ZnS:Cu/^6^LiF. The evaluated mixtures can be split into two categories. Four scintillation screens are ^6^LiF-based while two are Gd-based. A detailed description of the material properties is given in Section 2.1. The scintillation screen mixtures listed in Table 1 were provided by *RC Tritec AG* [13] and were chosen because their scintillating material is expected to show a fast decay. To characterize each scintillation screen, we have measured its light yield during the first few seconds of irradiation with neutrons, as well as after the end of irradiation.

For ZnS:Cu/^6^LiF, we additionally analyzed the change of light yield for exposure to different neutron flux levels as well as for different scintillation screen temperatures. It must be noted that using only the light yield does not give a full characterization of the used scintillation screens. The conversion efficiency as well as the neutron detection efficiency were not studied in detail. Although we did not performed a study on the inherent limitations in spatial resolution of the scintillation screens, we do expect a behavior similar to standard scintillation screens i.e., the spatial resolution is primarily influenced by the thickness of the scintillation screen.

In this publication, we give a phenomenological description and analysis of the light yield change of the scintillation screens due to sudden flux changes. Here, the interesting process is not the absorption of the neutron by the scintillation screen, but the excitation of the scintillating material by the absorption products. In the ^6^LiF-based scintillation screens the excitation radiation is the same, hence we focus on those screens when comparing the light yield changes.

## 2. Materials and Methods

### 2.1. Evaluated Scintillation Screens

All six scintillation screen mixtures were coated onto a 30 mm×30 mm aluminum substrate plate with a thickness of 1.5
mm. The scintillation material coating has shown some slight damages close to the outer edge of the substrate plate. These damaged areas are not visible in the experiment, as the mounting of the scintillation screens limits the visible area to 28 mm×28 mm. We split the scintillation screens into two groups, the ^6^LiF-based screens and the Gd-based screens. To reduce the required measurement time, we measured the scintillation screens of one group simultaneously. The mixing ratio by weight, the scintillation screen mixture thickness and the compounds of the evaluated screens are given in Table 1.

To simplify the discussion of the results we abbreviate the full chemical composition of the scintillation screen mixtures shown in the second column of Table 1 to the most relevant information for the discussion i.e., ZnS:Cu/^6^LiF is abbreviated as ZnS. The full list of abbreviations is given in the first column of Table 1. In case of the four ^6^LiF-based screens (ZnS, ZnCd, YAG and ZnO), compound 1 is the scintillation material responsible for the light emission. Compound 2 acts as the converter material, converting neutrons into ionizing radiation, activating compound 1. In case of the two Gd-based screens (Li-Gadox and Gadox) such a distinction is not straightforward, as compound 1 acts as both the converter as well as the scintillation material. The different scintillating materials were selected because their luminous decay-time to 10% from optical excitation is below 0.5
ms, according to *RC Tritec AG*. The scintillation screen thickness of 125 μm for the ^6^LiF-based screens was chosen as a compromise between detection efficiency and spatial resolution during high frame rate neutron imaging. Gd-based screens are generally thinner due to the stronger absorption of neutrons in Gd [14], as well as the self absorption of their emitted light.

### 2.2. General Experimental Parameters

The experiments were performed in the first measurement chamber of the ANTARES imaging beamline at the *Research Neutron Source Heinz Maier–Leibnitz (FRM II)* [15,16]. The polychromatic spectrum of ANTARES, a mixed thermal and cold neutron spectrum with its peak at 1.6Å, was used for the experiment. The spectrum was measured by Tremsin et al. [2]. The detection plane of the detector was set L=8.9 m downstream of the pinhole. For the experiments with a neutron flux of 2.5 $\times 10^{8}$ n cm^−2^ s^−1^ a pinhole diameter (D) of 35.6
mm was used. In case of a neutron flux of 6.25 $\times 10^{7}$ n cm^−2^ s^−1^ a pinhole diameter of 17.8
mm was used. This resulted in an L/D ratio of 250 and 500, respectively. As a detector an *ANDOR Neo 5.5 sCMOS* [17] camera with a *ZEISS Milvus 2/100M* [18] lens was chosen. The detector had a field of view of 71 mm×76 mm and an effective pixel size of 33 μm × 33 μm. The camera was set to a frame rate of 1 Hz with an exposure time of 1 s per image. Measurements of the light yield at an increased temperature were performed at 70 °C, which is significantly higher than room temperature while still being well below the maximum temperature (100 °C) of the organic binder [13] used in the scintillation screens. Higher temperatures may cause degradation of the organic binder and influence the measured light yield.

### 2.3. Evaluation of the Light Yield

To evaluate the light yield of the tested scintillation screens, their emitted light was captured using the detector system described in Section 2.2. To shut off the neutron beam quickly, the fast shutter system of the beamline was used. This shutter consists of an aluminum container of 1 cm thickness filled with B4C powder, which can be moved into the neutron beam in roughly 0.1
s. This system suppresses more than 99.99% of the thermal neutron flux. To compensate for a signal offset by the remaining neutrons as well as γ-flux, dark current (DC) images were taken while the beam has been shut off only by the fast shutter system. For the comparison of the light yield discussed in Section 3.1 the scintillation screens were first brought to an equilibrium state by irradiating them with a neutron flux of 2.5 $\times 10^{8}$ n cm^−2^ s^−1^ for 200 s. Then, 60 images (Sample) with an exposure time of 1 s were taken with the same neutron flux. For the offset correction 60 DC images were taken to compensate for thermal offset as well as the above mentioned offset caused by background radiation passing the fast shutter system. To calculate the relative light yield presented in Figure 1, an intensity value (I) was calculated for each scintillation screen using the following scheme: (1)$$I\left(m,n\right)={\mathrm{Sample}}_{\mathrm{pixmean}}\left(m,n\right)-{\mathrm{DC}}_{\mathrm{pixmean}}\left(m,n\right)$$
(2)$$I=I_{\mathrm{mean}}$$

Here, pixmean indicates a pixel-wise mean over the acquired images and mean indicates averaging over all pixels (m,n) of a single image. For the evaluation of the light yield in Section 3.2 and Section 3.3 the exposure time was set to 1 s.

### 2.4. Evaluation of the Absorption Data

To calculate the absorption of neutrons in the scintillation screens, radiographs of the screens were taken using a standard neutron scintillation screen (ZnS:Cu/^6^LiF) with a thickness of 100 μm. The measurement process is similar to the the one described in [19]. Three sample images of the screens (Sample) with an exposure time of 1 s were taken. To compensate for neutron beam and scintillation screen inhomogeneities, three reference images (Ref), without the samples in the neutron beam, were taken at an exposure time of 1 s. To correct for the effect of thermal offset, as well as camera graylevel offset three DC images with 1 s exposure time were taken. To analyze the absorption of the tested scintillation screens, the neutron transmission (TI) at every pixel (m,n) through the scintillation screens was calculated as: (3)$$\mathrm{TI}\left(m,n\right)=\frac{{\mathrm{Sample}}_{\mathrm{pixmed}}\left(m,n\right)-{\mathrm{DC}}_{\mathrm{pixmed}}\left(m,n\right)}{{\mathrm{Ref}}_{\mathrm{pixmed}}\left(m,n\right)-{\mathrm{DC}}_{\mathrm{pixmed}}\left(m,n\right)}$$
where pixmed indicates a pixelwise median over the acquired images. To increase statistics of the absorption (Abs) the mean value over an area of 20 mm×20 mm in the center of each scintillation screen of the transmission was used: (4)$$\mathrm{Abs}=1-{\mathrm{TI}}_{\mathrm{mean}}$$

## 3. Results

### 3.1. Comparison of the Light Yield

In Figure 1, the resulting light yield for every scintillation screen mixture normalized to the light yield of ZnS as well as the neutron absorption percentage of each screen is presented. The measurement and evaluation procedures for these results are explained in Section 2.3 and Section 2.4. Unless noted otherwise, all measurements were performed at a scintillation screen temperature of 25 °C. Li-Gadox (purple) and Gadox (brown) show high neutron absorptions of (67.3 ± 2.4)% and (68.3 ± 3.2)%, respectively. However, their light yield is comparably low with a normalized light yield of (5.8 ± 0.2)% and (3.8 ± 0.2)%, respectively. ZnO (orange) has an absorption of (19.6 ± 2.5)% with a normalized light yield of (0.1 ± 0.05)%. Due to the very low light yield, a quantitative evaluation of the time dependent light yield was been possible. Hence, no further light yield data of this scintillation screen are shown. YAG (blue) absorbs (12.7 ± 2.9)% of the incoming neutrons, while providing a normalized light yield of (9.1 ± 0.2)%. ZnCd (red) shows the highest normalized light yield of (63.3 ± 2.5)% with an absorption of (29.6 ± 1.9)%. As ZnS (green) was used to normalize the light yield of all other scintillation screens it has a normalized light yield value of (100 ± 1.8)%. It additionally shows an absorption of (11.7 ± 2.5)%.

### 3.2. Initial and Long Term Light Yield during Constant Neutron Irradiation

We performed a series of 1 s exposures for 1000 s to analyze the increase in light yield in the beginning of neutron irradiation (buildup). Time $t_{i}=0\;s$ denotes the first image with the neutron beam fully open. The screens have been irradiated with a neutron flux of 2.5 $\times 10^{8}$ n cm^−2^ s^−1^.

The resulting data are presented in Figure 2a. The light yield of every scintillation screen was normalized to its respective light yield at $t_{i}$.

The scintillation screens was measured in two groups, hence the light yield changes induced by a change in reactor flux are the same within one group of the scintillation screens. Therefore, we expect the distinct features present in all time-dependent light yield curves of a single group to be indicators of neutron flux variations caused by the reactor control. Such as the dip at t≈200 s or the oscillation starting at t≈700 s present for ZnS, ZnCd and YAG but not visible for Gadox and Li-Gadox. ZnCd, YAG, Gadox and Li-Gadox show no significant time-dependence over the range of 1000 s, apart from the previously mentioned fluctuations. In contrast, the relative light yield of YAG decreases roughly with a rate of 8 × 10^−5^s^−1^ to 92% until $t_{f}=1000\;s$. The relative light yield of ZnS first rises asymptotically until $t_{\mathrm{peak}}\approx 60\;s$ to 104.5% and then linearly decreases with a rate of ≈2.5 $\times 10^{-5}$ s^−1^ to 102% at $t_{f}$.

As ZnS is one of the most commonly used scintillation screen materials in neutron imaging, we additionally analyzed the buildup shown in Figure 2a under increased temperature as well as lower neutron flux. For these experiments we used the light yield of ZnCd, measured simultaneously, to remove the influence of the varying neutron flux. Hence, we first normalized all data to $t_{i}=0\;s$ and then normalized ZnS to ZnCd, as its buildup is below the fluctuations of the neutron flux (Figure 2a). In Figure 2b the resulting buildup curves of ZnS at a temperature of 70 °C and a flux of 2.5 $\times 10^{8}$ n cm^−2^ s^−1^ (orange), as well as the buildup curve at a temperature of 25 °C and a flux of 6.25 $\times 10^{7}$ n cm^−2^ s^−1^ (light orange) are compared to the buildup curve (green) at standard measurement conditions (2.5 $\times 10^{8}$ n cm^−2^ s^−1^; 25 °C) as used in Figure 2a. To compare the light yield curves taken at different neutron fluxes, the relative light yield is plotted versus the neutron fluence Φ. During the deposition of the first 0.5 $\times 10^{10}$ n cm^−2^, the increase in light yield of all three curves is the same. Afterwards, the two curves measured with different fluxes at 25 °C display good agreement. The maximum relative light yield of the curve at lower flux is at 103.5%, which is slightly lower than the 104% of the high flux curve. The two curves both reach their peak at $\Phi_{\mathrm{peak}}\approx$ 1.5 $\times 10^{10}$ n cm^−2^. In contrast, the buildup at 70 °C reaches 104.5% and is slightly delayed compared to the other two curves at $\Phi_{\mathrm{peak}}\approx$ 2.6 $\times 10^{10}$ n cm^−2^. We again observe a decay of the relative light yield at neutron fluences above $\Phi_{\mathrm{peak}}$. For the high flux curve, the relative light yield decreased with a rate of ≈2.1 $\times 10^{-3}$ % s^−1^. The relative light yield of the low flux curve and the curve at 70 °C decreased with a slower rate of ≈1.5 $\times 10^{-3}$ % s^−1^ and ≈1.3 $\times 10^{-3}$ % s^−1^, respectively.

To analyze the light yield change during constant neutron irradiation we have combined the light yield curves of different individual scans performed over the course of the whole experiment. The resulting long term light yield curves of ZnCd (red), ZnS (green) and YAG (blue) dependent on the total deposited neutron fluence are presented in Figure 3. Before explaining the intensity curves themselves, we first mention some peculiarities visible in these curves. In area A, the screens were exposed to a neutron fluence of 3 $\times 10^{11}$ n cm^−2^, but no measurements were performed. The dashed line B denotes a pause of 12 h, during which the screens have not been exposed to neutrons. This pause was used to measure the two Gd-based scintillation screens. The scintillation screens were heated to T = 70 °C during the fluence deposition in area C. The relative light yield decrease of YAG (blue) per deposited neutron fluence lessens at higher total absorbed neutron fluence, while the heating of the scintillation screen has a negligible effect on the light yield. At the end of the experiment, the light yield has decreased to 57%. The light yield of ZnS (green) shows a decrease to 95% at a total deposited fluence of 2.38 $\times 10^{12}$ n cm^−2^. After heating the scintillation screen to a temperature of 70 °C, the light yield drops to 91%. At the end of the experiment the light yield has dropped to 87%. The light yield curve of ZnS shows dips corresponding to the buildup of light yield in the beginning of irradiation for each of the individual measurements. The light yield of ZnCd (red) decreases to 98% until a total neutron fluence of 2.38 $\times 10^{12}$ n cm^−2^ has been deposited. After increasing the scintillation screen temperature to 70 °C, the light yield drops to 91% but shows no further decrease until the end of the experiment. All scintillation screens show a slight increase of light yield after the measurement pause of 12 h.

### 3.3. Decay Characteristics

Moreover, we evaluated the light yield of the scintillation screens after the end of irradiation i.e., afterglow. To reach a stable equilibrium before shutting off the neutron beam, the scintillation screens were first brought to saturation by continuous irradiation with a flux of 2.5 $\times 10^{8}$ n cm^−2^ s^−1^ over a duration of at least 600 s. Afterwards, the neutron beam was shut off and the resulting light yield was measured. To make all scintillation screens comparable, the light yield was normalized to the light yield $t_{i}=0\;s$ just before the shut down of the neutron beam. In Table 2, the light yield of all scintillation screens at $t_{i}$ normalized to ZnS is given.

The images were taken continuously with an exposure time of $t_{\exp}=1\;s$. Figure 4a presents the resulting time-dependent relative light yield. During the exposure at t=1 s the neutron beam was shut off, hence this value does not provide reliable information. As a consequence the data points at t=1 s were removed from the graphs. We note that the relative light yields at t=2 s differ within an order of magnitude. ZnS (green) shows the highest value and ZnCd (red) the lowest, with 1.98% and 0.14%, respectively. The relative light yields of YAG (blue), Gadox (brown) and Li-Gadox (purple) lie between these two extrema with 0.20%, 0.32% and 0.48%, respectively. At t=40 s the relative light yield of ZnS, Gadox and Li-Gadox has decreased to roughly 0.1%. The relative light yields of ZnCd and YAG have decreased to 0.03% and 0.07%, respectively.

In Figure 4b the decay curves of ZnCd, ZnS and YAG at a flux of 2.5 $\times 10^{8}$ n cm^−2^ s^−1^ and a temperature of 25 °C (round markers) are compared to the decay at a higher temperature of 70 °C, without changing the flux (diamond markers) and a lower flux of 6.25 $\times 10^{7}$ n cm^−2^ s^−1^ and keeping the temperature at 25 °C (square markers). The images were taken continuously with an exposure time of $t_{\exp}=1\;s$.

For ZnCd and YAG, the curves of the three different measurement conditions are in very good agreement, showing that there is no influence of the measurement conditions on the decay. In contrast, the decay of ZnS is linked to the specific measurement conditions. Using a flux of 2.5 $\times 10^{8}$ n cm^−2^ s^−1^ to irradiate the scintillator, the relative light yield decreased to 0.06% at t=40 s after the end of irradiation when the scintillator was heated to 70 °C, while the relative light yield at 25 °C decreased to 0.13%. In contrast, the light yield with a flux of 6.25 $\times 10^{7}$ n cm^−2^ s^−1^ decreased to 0.22%. The greatly varying relative noise levels are due to the vastly different absolute light levels of the scintillators while the noise in the detector system is fairly constant. Hence, the scintillators with a low light yield, e.g., YAG show a worse signal to noise ratio than the light intensive scintillators (e.g., ZnS).

## 4. Discussion and Conclusions

In this paper, we presented our findings on five novel scintillation screen materials (ZnCd, YAG, ZnO, Li-Gadox, Gadox) shown in Table 1 as well as compared them to ZnS, which is one of the most common scintillation materials used in neutron imaging. The light yield of the ^6^LiF-based scintillator screens tends to be generally higher than the light yield of the Gd-based scintillators, as can be seen in Figure 1. ZnS shows the highest light yield and has been used as a reference for the other scintillators. Li-Gadox and Gadox show a high neutron absorption ((68.3 ± 3.2)% and (67.3 ± 2.4)%) but a low light yield ((5.8 ± 0.2)% and (3.8 ± 0.2)%) compared to the light yield of ZnS. ZnCd shows the second highest normalized light yield with (63.3 ± 2.5)%, while YAG has a normalized light yield of only (9.1 ± 0.2)%. The neutron absorption of ZnCd is (29.6 ± 1.9)%, whereas YAG absorbs (12.7 ± 2.9)% of the incoming neutrons. The higher absorption in the thinner Gd-based screens is expected, as Gd has a much higher neutron absorption cross section (49700 b) than ^6^Li ( 940 b) [14]. The low relative light yield of the Gd-based screens can be explained by the different daughter products of the conversion reactions of ^6^Li and Gd. Gd produces a conversion electron with an energy of 29 keV to 191keV when capturing a neutron, whereas ^6^Li produces an α-particle with 2055 keV and a ^3^H-particle with 2727 keV. The lower amount of scintillation material, due to the smaller thickness of the Gd-based screens, as well as the lighter and less energetic particles generated by the neutron absorption leads to less light being emitted by Gd-based screens compared to ^6^LiF-based screens. In contrast, the heavier and more energetic particles produced by ^6^Li as well as the higher scintillation screen thickness leads to more energy being deposited in the scintillation material, typically causing higher light yields. Contrary to this trend, ZnO barely shows any measurable light yield and is therefore not suitable for neutron imaging.

Even though the basic scintillation mechanism of electron excitation is the same for all four scintillation materials in the ^6^LiF-based screens, the specific mechanisms governing the probability of light emission are influenced by the specific crystal structure and are beyond the scope of this paper. A more detailed view upon the underlying scintillation processes is given by Getkin et al. [20].

It should be noted that the higher neutron absorption of ZnCd does not necessarily indicate a higher detection probability of neutrons, as the Cd in this screen results in additional absorption. Although the absorption of neutrons in Cd produces γ-rays, we expect no significant contribution to the light yield caused by γ-rays. Typically, ZnS-based scintillation screens show low sensitivity to γ-rays. As Spowart [21] suggests that 10^3^ photons are required to produce the same light yield as a single neutron.

Considering the buildup of light yield in the beginning of irradiation (first 1000 s) shows that Li-Gadox, Gadox and ZnCd do not have an appreciable buildup of light yield, whereas ZnS displays an increase of light yield to 104%. YAG also shows no buildup, but suffers from decreasing light yield. The buildup curve of ZnS is thought to be a combination of three different processes. Typically, ZnS:Cu has intrinsic electron traps. In the beginning of irradiation a part of the electrons excited by the neutron capture process show no luminescence, but are captured by these traps. The electrons in the traps may decay later, showing delayed scintillation. Continuing neutron radiation causes an equilibrium between the charging and decaying of the traps, as well as the direct fluorescence [22]. This initial equilibrium is reached roughly 60 s after the beginning of irradiation, see Figure 2a. Afterwards, ZnS shows a continuing decrease of light yield due to its relatively poor radiation hardness. Furthermore, the damage to the crystal structure caused by ionizing radiation also leads to the creation of color centers i.e., defects which act as electron or hole traps further reducing the light yield [20,23,24,25]. The reduction in light yield continues during the whole experiment, as seen in Figure 3. Over the 12 h break in the long term measurements indicated by the dashed line B in Figure 3, the color centers and defects in the crystal structure are partially annealed, which leads to a slightly increased light yield [26]. An increase in temperature causes thermal quenching for ZnS:Cu, reducing the light yield, while the trend of decreasing light yield due to radiation damage continues [27].

The neutron fluence dependent buildup of ZnS in Figure 2b using different neutron fluxes and scintillation screen temperatures shows a slightly larger maximum relative light yield, i.e., a stronger buildup, for the high neutron flux compared to the low flux. Simultaneously, the relative light yield decreases faster with rising neutron fluence in the case of higher neutron flux. This indicates that the filling of traps and the creation of defects in the crystal structure is primarily neutron fluence dependent while the decay of traps is primarily time dependent. Hence, a higher neutron flux causes more traps to be filled simultaneously, increasing the peak buildup. At the same time, increasing the neutron flux leads to increased defect creation, indicated by the faster decline of relative light yield during irradiation with high flux [20,23].

The increased relative buildup of the heated scintillation screen is presumably caused by the fact, that the number of intrinsic traps stays the same over an increase in temperature, while the direct fluorescence is quenched. However, we expect the life time of the electrons in the traps to be reduced as their decay may be thermally induced. Hence, the general shapes of the curves are similar, but the absolute buildup values originate from different starting conditions and are therefore not comparable. Heating of the scintillation screen seems to lead to a slightly slower decrease of light yield (≈$1.3\times 10^{-5}$ s^−1^) compared to the scintillation screen at room temperature (≈$2.1\times 10^{-5}$ s^−1^).

The light yield curve of ZnCd during constant neutron irradiation shows only a small decrease of light yield, indicating an increased radiation hardness. Here, the Cd in ZnCd is suggested to be a stabilizing agent reducing radiation damage by quenching the luminescence caused by the luminescence centers [20]. The measurement break of 12 h also leads to a small increase of light yield again indicating an annealing of crystal defects. The effect of quenching at higher scintillation screen temperatures is slightly stronger for ZnCd compared to ZnS.

The strong decrease of light yield in YAG is surprising, as YAG has been reported to be stable with respect to radiation [28]. It needs to be noted that the radiation hardness test in [28] was performed using a 3 MeV proton beam, hence the results may not be directly comparable. YAG also shows no thermal quenching, which may be a result of a different crystal structure compared to the other two scintillation screens. Similar to the other screens YAG shows an increased light yield after the 12 h resting period.

After the end of irradiation ZnS shows an order of magnitude higher afterglow than ZnCd, while the afterglow of YAG, Li-Gadox and Gadox lies in the range between these two extremes. We note that the decay of the scintillation screens does not follow a single exponential decay, but is a superposition of various exponential decays. While the data curves may be fitted with multiple exponential functions, the resulting parameters would give no further information. Hence, no fitting of the data curves was performed. One should also note that due to the low absolute light yield of YAG, Li-Gadox and Gadox, the noise in the data becomes more pronounced. This noise is caused by γ-radiation hitting the detector as well as thermal noise in the camera. Incidentally, this noise may also be the reason for the flattening of the curves, as the constant noise may hide a slowly decaying light output. The leveling off of the curves occurs at different relative levels as the light yield before shut off of the neutron beam was different for every scintillation screen, while the noise level in the detector system was constant for every scintillation screen. This may also explain the crossing of the decay curve of ZnS with the curve of Li-Gadox at t=30 s. The high afterglow of ZnS probably originates from radiation induced luminescence centers. As “de-trapping” is thermally induced, the increase of temperature leads to a faster decay. Furthermore the thermally induced annealing process reduces the amount of induced luminescence centers and hence reduces the afterglow.

As noted before, previous studies mostly covered the response of scintillation screens in the μs-range [6,7]. Furthermore, these studies were performed at low flux sources (≈10^6^ n cm^−2^ s^−1^). Hence, the long decay time we evaluated may be hidden in the measurement noise of these studies, similar to our experiences with YAG, Li-Gadox and Gadox.

One important property of neutron scintillation screens, their spatial resolution, has not been touched upon in this study. However, we expect the scintillation screens to follow roughly the same trend of spatial resolution versus thickness as observed in standard scintillation screens offered by *RC Tritec AG* [13]. Comparing the five new scintillation screen mixtures, the ZnCd mixture has the highest light yield, while the Gd-based screens have, with the exception of the ^6^LiF-based ZnO screen, the lowest light yield. While the decay time of Li-Gadox and Gadox is slightly faster than that of ZnS, their low light yield per detected neutron increases the Poisson noise in the camera system [1]. While this property can be overcome with increased exposure times, this is not feasible for high frame rate neutron imaging. While Gd-based scintillation screens are not suitable for high frame rate imaging, their high detection probability at low thicknesses makes them a good choice for high spatial resolution imaging. The non-existent buildup and low afterglow of YAG would make it well suited for high frame rate neutron imaging. In contrast, the continuously decreasing light yield as well as the generally low light yield compared to other ^6^LiF-based scintillation screens make this mixture unattractive for neutron imaging. ZnCd has 63% of the light yield of the standard ZnS mixture. The buildup is in the range of the neutron flux fluctuation of the reactor. Furthermore, for ZnCd we measured the fastest decay of afterglow of all evaluated scintillation screens. ZnS decays to 1.98% of its previous light yield in 1 s, while ZnCd decays to 0.14% in the same time. An advantage of ZnCd compared to other fast decaying scintillation screens [29,30] is the high light yield per detected neutron, which is much higher than observed in Gd-based scintillation screens or in Li-glass scintillation screens [1]. This is advantageous as a higher photon count decreases the Poisson noise in the camera system. A further advantage of ZnCd compared to Gd-based scintillation screens is the good γ discrimination. Hence, the combination of fast buildup and decay, as well as high overall light yield with a high light yield stability makes ZnCd a good alternative to ZnS especially for high frame rate neutron imaging.

Comparing our result for the decay time of ZnS ( 2.5±0.5
s) to 1% to previously reported decay times of 85 μs to 10% [5,29] there is a large discrepancy. While one explanation of the differences in decay time could be attributed to differences in the doped ZnS scintillation compound, a difference on the order of more than 1 × 10^5^ seems unlikely. As we have noted, the decay of the light yield does not follow a single exponential decay, hence it may be that the ZnS screen we have measured also decayed to 10% in microseconds, which was however not accessible with the time resolution of our measurements. Another explanation for the different results may be down to differences in the excitation of the scintillation material. In [5], it seems that the provided decay curve of a ZnS/^6^LiF scintillation screen was measured using optical light with 337 nm wavelength to excite the scintillation screen. For other scintillation screen materials it has been shown previously that the type of exciting radiation influences the decay time [31]. Hence, we expect also for ZnS/^6^LiF that the measured decay time depends on the excitation radiation. As such, great care must be taken when comparing different scintillation screen decay time measurements, as the type and the intensity of the excitation radiation must be taken into account. Furthermore, as we have also seen in our experiments, thermal noise inherent to the detector system may hide light yield decay.

In conclusion, a Gd-based screen is best suited for high spatial resolution neutron imaging, while ZnCd is most suited for high frame rate neutron imaging. To further understand the mechanisms governing the response of neutron scintillation screens, especially ZnS, to varying flux levels, it is planned to perform buildup and decay measurements at different timescales at high flux sources. ZnCd in particular will remove limitations in neutron imaging frame rates, as the fast decay reduces the ghosting effect i.e., information of one frame in another. This will also allow the acquisition of more quantitative results as one systematic error in the data can be eliminated.

## Figures and Tables

**Figure 1 jimaging-06-00134-f001:**
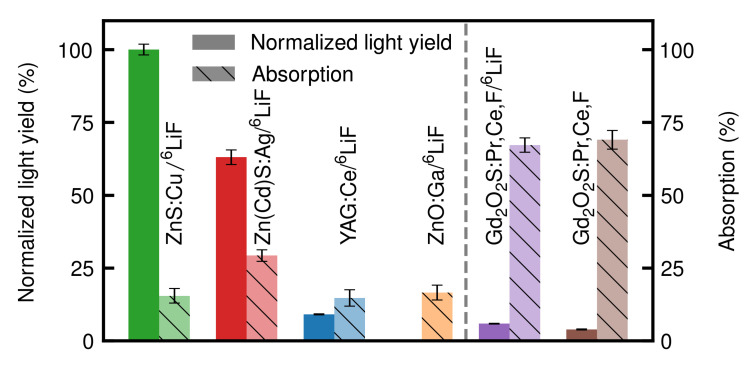
Neutron absorption of the six scintillation screen mixtures (hatched bars) and the resulting light yield (bars). The scintillation screens are split into two categories, the ^6^LiF-based screens on left of the dashed line and the Gd-based screens on the right. The scintillation screens in one group were irradiated simultaneously.

**Figure 2 jimaging-06-00134-f002:**
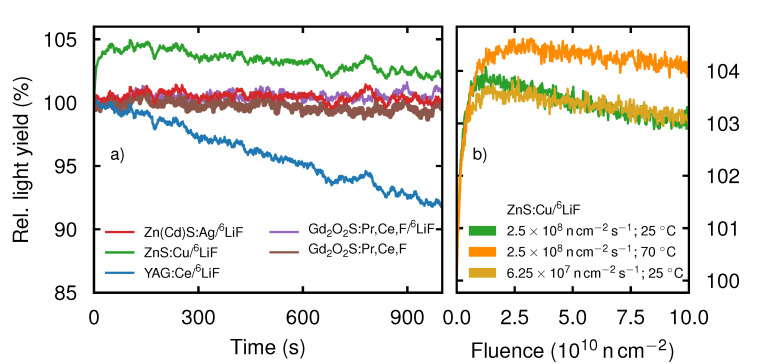
(**a**) Time-dependent light yield of the five scintillation screens during the first 1000 s of neutron irradiation relative to their respective light yield at time $t_{i}=0\;s$. (**b**) Change of the fluence-dependent light yield of ZnS upon change of the neutron flux or the temperature of the scintillation screen, respectively. The curves were adjusted for time-dependent neutron flux variations and normalized to the respective light yield at $t_{i}=0\;s$.

**Figure 3 jimaging-06-00134-f003:**
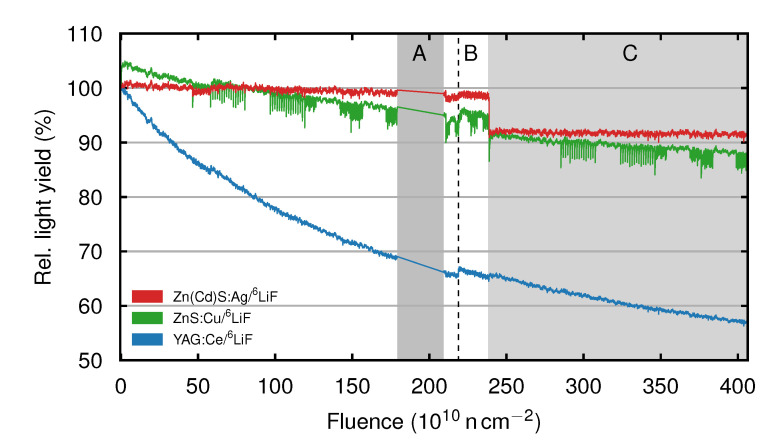
Light yield of ZnS (green), ZnCd (red) and YAG (blue) relative to their respective light yield at $t_{i}=0\;s$ dependent on the total deposited neutron fluence. In area A, the screens were exposed to a total neutron fluence of 3 $\times 10^{11}$ n cm^−2^, but no measurements were performed. The dashed line B denotes a pause of 12 h were the ^6^LiF-based screens were not exposed to neutrons. During this pause, we measured the Gd-based scintillation screens. The scintillation screens were heated to 70 °C during the fluence deposition in area C.

**Figure 4 jimaging-06-00134-f004:**
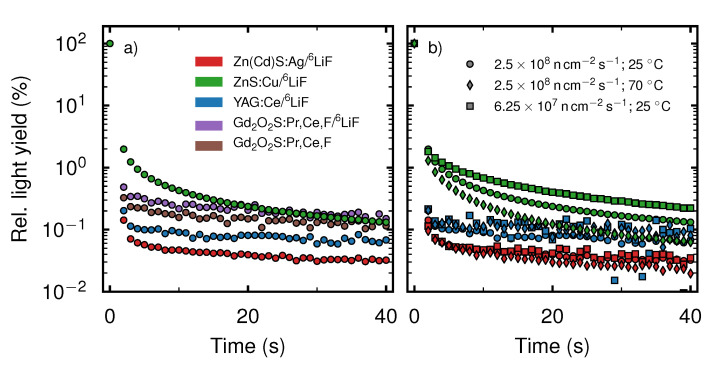
(**a**) Relative light yield decay of ZnS (green), ZnCd (red), YAG (blue), Gadox (brown) and Li-Gadox (purple) during the first 40 s after the end of neutron irradiation. All data are normalized to the last irradiated image at $t_{i}=0\;s$. (**b**) Change of the light yield of ZnS (green), ZnCd (red) and YAG (blue) during the first 40 s after shutting off the neutron beam. All data are normalized to the last irradiated image at $t_{i}=0\;s$. Round markers denote a measurement under standard conditions, whereas diamond markers indicate a higher temperature and square markers a lower neutron flux.

**Table 1 jimaging-06-00134-t001:** Physical and chemical properties of the evaluated scintillation screens.

Abbreviation	Item	Compound 1	Compound 2	Ratio	Thickness (μm)	Area (mm^2^)
ZnS	ZnS:Cu/^6^LiF	ZnS:Cu	^6^LiF	2:1	125	28×28
ZnCd	Zn(Cd)S:Ag/^6^LiF	Zn(Cd)S:Ag	^6^LiF	3:1	125	28×28
YAG	YAG:Ce/^6^LiF	YAG:Ce	^6^LiF	2:1	125	28×28
ZnO	ZnO:Ga/^6^LiF	ZnO:Ga	^6^LiF	2:1	125	28×28
Li-Gadox	Gd_2_O_2_S:Pr,Ce,F/^6^LiF	Gd_2_O_2_S:Pr,Ce,F	^6^LiF	4:1	40	28×28
Gadox	Gd_2_O_2_S:Pr,Ce,F	Gd_2_O_2_S:Pr,Ce,F	none	n.a.	25	28×28

**Table 2 jimaging-06-00134-t002:** Normalized light yield at $t_{i}$ of all light yield decay measurements presented in Figure 4.

	ZnS	ZnCd	YAG	Li-Gadox	Gadox
Neutron flux (2.5 × 10^8^ n cm^−2^s^−1^)	100%	64.7%	8.3%	5.9%	3.9%
Temperature (25 °C)					
Neutron flux (6.25 × 10^7^ n cm^−2^s^−1^)	24.7%	16.1%	2.0%	–	–
Temperature (25 °C)					
Neutron flux (2.5 × 10^8^ n cm^−2^s^−1^)	89.5%	59.7%	6.0%	–	–
Temperature (70 °C)					

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
