# Peer review of "Light Yield Response of Neutron Scintillation Screens to Sudden Flux Changes"

_2313-433X, 2020, doi:10.3390/jimaging6120134_

Round 1

Reviewer 1 Report

This manuscript presents the results of light yield measurements of five novel neutron scintillators. These results are of interest to and valuable for the neutron imaging community, especially those interested in high frame rate imaging and neutron scintillator screen development.

Overall, this is a nice paper and the experimental work described in this article is sound and useful. However, (as described in further detail below) more discussion on the physics of the scintillator screens is necessary to provide a complete story and emphasize the significance of these results and their applications. 

Line 11: "Nondestructive testing of cultural and technological objects is a key feature of neutron imaging." Consider rewording this sentence to reduce confusion. The nondestructive testing of objects is the key feature of neutron imaging, not the testing of cultural or technological objects.

Line 23: Can the authors provide a reference for the claim "...one of the standard scintillator screen mixtures (ZnS:Cu/6LiF) shows a relatively slow decay time of approx. 2.5 s to 1% of the previous light yield at the end of the illumination with neutrons."? Most commonly the reported 10% decay time for ZnS:Cu/6LiF is 85 μs and Figure 8 in the reference shown below this comment does not seem to agree with the authors' assertion that 1% of light yield occurs at 2.5 s.

Schillinger, B., et al. "Detection systems for short-time stroboscopic neutron imaging and measurements on a rotating engine." Nuclear Instruments and Methods in Physics Research Section A: Accelerators, Spectrometers, Detectors and Associated Equipment 542.1-3 (2005): 142-147.

Line 37: "A further challenge is posed by different scintillation screen thicknesses, as the Gd-based scintillation screens have material related a smaller thickness." A word (or more) appears to be missing from this sentence as it does not currently make sense as written.

Can the authors also elaborate on why they chose the specific thicknesses of scintillator screens? Calculations of and a discussion on the mean free path of a neutron in the scintillator screen materials may help provide justification for this. 

Table 1: Were the substrates the same for each scintillator screen? The authors should describe the substrate material because different substrates (reflective, matte black, etc.) can directly impact light yield. If different, the substrate material should be added to this table. If they are all the same it would be acceptable to mention the material in the preceding paragraph.

This table lists "Compound 1" and "Compound 2," however it may be more accurate (with the exception of Gd2O2S:Pr,Ce,F) to label "Compound 1" as "Converter Material" and "Compound 2" as "Scintillator Material" as this gives more description on the purpose of the compounds in the scintillator.

Is the area reported in this table of the active scintillator material or of the substrate itself? Please clarify as this has implications for field of view, scintillator screen mounting, etc.

Line 42: "First we analyzed the absolute light yield of the six scintillation screen compounds." This is not an absolute light yield measurement, but a relative light yield measurement as the measured scintillators are compared to one another. The authors must take care in terminology here because absolute light yield is a specific measurement that determines the number of photons a scintillator releases per quanta of energy absorbed. An absolute light yield measurement for neutron scintillators has remained elusive to researchers because neutron sources generally give a spectrum of neutron energies, as well as gamma rays and other ionizing radiation. This makes it difficult to determine exactly how much energy is absorbed by a neutron scintillator. For a full discussion on performing an absolute light yield measurement (with gamma-ray radiation) please see "M. Moszynski, M. Kapusta, M. Mayhugh, D. Wolski and S. O. Flyckt, "Absolute light output of scintillators," in IEEE Transactions on Nuclear Science, vol. 44, no. 3, pp. 1052-1061, June 1997, doi: 10.1109/23.603803."

Figure 2: The authors need to check the formatting of this figure. When printed, the hatched bars (absorption data) are not shown, although they appear in the e-copy. Figure file type may need to be changed. 

Additionally, normalizing the light yield data (y-axis on the left side of the figure) and reporting it as a percentage of the standard ZnS:Cu/6LiF light output makes this more valuable to the reader then the currently used arbitrary units. Later on in the discussion portion of the text it is not helpful to read that a certain scintillator has a light yield of 1, while another has a 1019 because the reader does not have a good concept of scale. Reporting as a percentage relative to a standard would be much more useful. The absorption data is reported as a percentage in the text (lines 53, 60, and 63), so changing the right-hand y-axis from relative units to percentage would help with consistency. 

Line 52: This section needs to be improved with a more thorough discussion on why different compounds have varying light yields. The authors are correct to mention the lower cross section of 6Li vs Gd being responsible for decreased neutron absorption, however these values should be provided. Additionally, the daughter products of these converter materials need to be discussed. For example, the higher light yield of 6Li is due to heavier daughter products that have a higher energy (relative to the Gd internal conversion electron) and therefore deposits energy into the scintillator screen. This is in contrast to the Gd based screens, which are generally thinner and allow some of the high energy daughter products to easily escape the screen without depositing their energy in the scintillator material, creating a lower light output. A detailed discussion on the converter daughter products and their interactions with the scintillator material is necessary not only to give a more complete understanding on the performance of these novel neutron scintillators, but it also improves the scientific value of this article.  

Line 57: "...while their light yield varies strongly." Why does the light yield of the four 6LiF compounds vary widely? The authors should provide details about the mechanisms of the scintillator materials that describe why these vary widely.

Line 69: "Initial and long term light yield during permanent neutron radiation." "Permanent" may not be the correct term here as it implies something lasts forever and these samples were not irradiated for a particularly lengthy period of time. "Constant irradiation" may better fit because the experiments described in this section appear to cover a constant irradiation during a time interval of approximately 24 hours. 

Figure 2b: The grey line by ZnS:Cu/6LiF should be removed. The text makes it clear that 2b is showing the performance of ZnS:Cu/6LiF is shown in this figure, but the grey line makes this confusing, especially because there is no grey line data in the plot. It would be best to either remove the grey line, remove the grey line and ZnS:Cu/6LiF label, or place the ZnS:Cu/6LiF label as a title.

Line 110: "The dashed line B denotes a pause of 12 h were the screens were not exposed to neutrons." Please give a reason for the pause. Was it due to operational considerations or an experimental desire to observe scintillator screen recovery.

Line 115: "The light yield of ZnS:Cu/6LiF (green) shows a medium decrease to 95%..." The word "medium" is confusing here because it is not quantitative and should be omitted. If the intent was to use the word "medium" to describe the change in light output, simply say "...shows a decrease to 95%."

Line 169: "In contrast to Gd-based screens the light yield of 6LiF-based screens tends to be higher." As previously mentioned, address why this is the case either here or in the introduction. A discussion of converter daughter products and their energies should explain the difference in light yields between scintillators with different converter materials.

Line 233: In the paragraph preceding this line, the authors give a nice discussion of various scintillator mechanisms and their impact on light output. Throughout the article, the authors correctly state light yield and light decay time as two important factors in scintillator selection. However, another important factor is neglected: spatial resolution. Spatial resolution measurements may be outside the scope of this study. However, can the authors offer any comments or speculation about how these various novel scintillators will differ in spatial resolution? Theoretically this may be due to daughter product ranges of the converters or light diffusivity in the various scintillator media.

Line 233: "In combination with its high light yield stability Zn(Cd)S:Ag/6LiF is a good alternative to ZnS:Cu/6LiF especially for high frame rate neutron imaging." This sentence is a little confusing. The high light yield stability, in combination with what, makes Zn(Cd)S:Ag/6LiF a good alternative to ZnS:Cu/6LiF. 

Semantics aside, more information needs to be given to justify what makes Zn(Cd)S:Ag/6LiF a good alternative to ZnS:Cu/6LiF. In line 23, the authors mention that "ZnS:Cu/6LiF shows a relatively slow decay time of approx. 2.5 s to 1 % of the previous light yield after the end of illumination with neutrons." How does Zn(Cd)S:Ag/6LiF compare? How long does it take for Zn(Cd)S:Ag/6LiF to decay to 1%? Having this direct comparison would help explain why Zn(Cd)S:Ag/6LiF is a good alternative to ZnS:Cu/6LiF. Additionally, the authors mention previous studies on time-resolved imaging in the sub-second regime (references 2-4). Is the afterglow decay of Zn(Cd)S:Ag/6LiF in the sub-second regime? If not, how does Zn(Cd)S:Ag/6LiF offer an advantage over scintillators with a quick decay time (neutron scintillators can have a decay time on the order of microseconds). See the following for examples of neutron scintillator screens with μs decay times:

"Hussey, Daniel S., et al. "Neutron imaging detector with 2 μm spatial resolution based on event reconstruction of neutron capture in gadolinium oxysulfide scintillators." Nuclear Instruments and Methods in Physics Research Section A: Accelerators, Spectrometers, Detectors and Associated Equipment 866 (2017): 9-12."

"Hillenbach, A., et al. "High flux neutron imaging for high-speed radiography, dynamic tomography and strongly absorbing materials." Nuclear Instruments and Methods in Physics Research Section A: Accelerators, Spectrometers, Detectors and Associated Equipment 542.1-3 (2005): 116-122." 

Reference 7 from the author's manuscript lists Gd3Al2Ga3O12:Ce as having a ~30 µs decay time.

Line 261: "...first the dark current was subtracted from both data and reference images." What is the "reference image" and what is the "data" image? Section 4.2 is unclear and needs more detailed explanation of the methodology used to obtain the data. Was a radiograph taken of the novel scintillators with a standard screen to measure neutron absorption, as described in Section 4.1 of "Chuirazzi, William C., and Aaron E. Craft. "Measuring Thickness-Dependent Relative Light Yield and Detection Efficiency of Scintillator Screens." Journal of Imaging 6.7 (2020): 56."? Or was a different procedure utilized? Please give more description or cite an appropriate reference that describes the procedure.

This is an interesting body of work that will benefit from a deeper discussion of scintillator physics such as daughter products, mean free path of neutrons in scintillator compounds, etc. This project also seems well positioned for future work whether that be using the top performing screen to image samples, quantifying spatial resolution of the screens, observing light yield recovery by heating screens/annealing, or designing another iteration of screens based on these results. Nicely done.

Reviewer 2 Report

This paper provides useful information for the neutron imaging scientists who develop new instruments and techniques. The data measured in this study are useful and the paper should be published after some revisions. 

The paper style format does not follow the Style Guide for this publication. The template provides the styles needed in a user friendly format.

The Introduction states, "...one of the standard scintillation screen mixtures
(ZnS:Cu/6LiF) shows a relatively slow decay time of approx. 2.5 s to 1% of the previous light yield after the end of illumination with neutrons." Is this measured in the present work? If not, then the authors should provide a reference.

The method should be described before results can be discussed. This paper moves from the Introduction directly into Results without discussing the measurement procedure. Is the neutron flux cold or thermal? Even such a basic question is not yet understood by the reader. 

Regarding Table 1: Please check the mix ratios of the screens. The table seems to be the opposite ratios of the screens that RC Tritec typically sells. The table includes ZnS:Cu/6LiF with a 1:2 mixture and Zn(Cd)S:Ag/6LiF with a 1:3 mixture (more 6LiF than ZnS). However, RC Tritec previously sold such screens with a 2:1 and 3:1 mixture ratio, respectively (more ZnS than 6LiF). Also, please clarify if these mixture ratios are weight or molecular ratios.

Section 2.1 begins, "First we analyzed the absolute light yield of the six scintillation screen compounds." Absolute light yield should be given in photons per neutron absorbed, photons per MeV, or some other quantitative unit. However, Figure 1 reports light yield with arbitrary units, and the reader assumes this is relative light yield.

Figure 1 seems to have a formatting issue in the PDF version of the paper. The data labels are not aligned with the chart.

Please mention why this study focuses on room temperature and 70C. Why 70C and not other temperatures?

"Dose" in Figure 2b is more accurately termed as "neutron fluence," which is time integrated neutron flux and given in n/cm2). For example, "20 dose units" is simply "2*10^11 n/cm^2." Please change the "dose" unit to neutron fluence in both Figure 2, the subsequent discussion of Figure 2, and all other mention of "neutron dose". ["^" simply denotes a superscript in these review comments.]

The first mention of Figure 2 in the text occurs after Figure 2. The figure should appear after its first mention.

Typo: 238*10^110 n cm^-2. should be 238*10^10 n cm^-2. 

As a general note, 238*10^10 n cm^-2 should be written 2.38*10^12 n cm^-2 per scientific notation.

Please move Figure 3 closer to its first mention and before Section 2.3. As written, the paper discusses Figure 4 before presenting Figure 3 to the reader.

The first sentence of Section 2.3 states, "Moreover, we evaluated the light yield of the scintillation screens after the end of illumination i.e afterglow." "Illumination" refers to the emission of light, so measuring light yield after the end of emission of light is confusing. Please clarify if the end of "illumination" refers to the end of neutron beam exposure.

Referring to Section 2.3, how was the beam shut off so quickly? How much time is required to fully close the neutron beam. Please provide this clarifying information.

Much of the information in the captions of the Figures is reiterated in the text. The reviewer recommends removing duplicate information from the Figure captions, keeping the detailed discussion in the text.

The "Discussion and Conclusions" section says, "Gd2O2S:Pr,Ce,F/6LiF and Gd2O2S:Pr,Ce,F show a high neutron absorption (68% and 67%) and a low light yield (59 and 39)." Without any units, light yield values carry no meaning. Consider making the light yield into a "relative light yield" and normalize them to the maximum, thereby giving values in percent.

Do the authors know whether ZnS afterglow behavior varies between suppliers of ZnS, and between different batches of ZnS? Such variations have been discussed previously. Perhaps it is useful to mention that these decay behaviors are specific to the screens tested in this study, but that variations between manufacturers and batches is POSSIBLE.

The "Materials and Methods" section should be the second section in this paper immediately following the Introduction and before any results are reported.

The discussion in Section 4.2 should be explained using equations to make the process clear to the reader.

The results section states, "For absorption measurements, the scintillation screens were placed in front of a neutron imaging detector system and their transmission was measured. The evaluation procedure is also explained in Sec. 4.2." However, Section 4.2 does not explain this process with any clarity. Please describe more clearly how the absorption measurements were made, both the experimental setup and the subsequent data processing.

Overall, this paper provides valuable and useful information. It is definitely worth the researchers' time to rework the paper to better explain the methods used and clarify the results. If refined, this paper should be a highly citable contribution to the neutron imaging field. 

Reviewer 3 Report

Dear authors, 

the manuscript "Light yield response of neutron scintillation screens to sudden flux changes" contains interesting information, however in my view it needs improvements in order to be accepted for publication. Some general comments are:

  • the structure of the paper is not optimal, which makes it hard to read and to follow the main ideas and results. 
  • the abstract should point out key results
  • the Introduction must describe the state-of-the-art in the field and give appropriate references
  • the Results section contains parts of the Discussion and vice versa, the Results section should focus on presenting the results and they should be interpreted and discussed in the Discussion
  • some figure captions are too extensive, containing a lot of information that should be in the text. Please use captions to briefly describe figures and discuss the results in the text,
  • A section that must be extended is the Materials and Methods - in current form it is unacceptable. The authors must describe the experimental setup in sufficient details that the readers can follow. Also, it is the usual practice to place the Materials and Methods section after the Introduction and before the Results. 
  • Language is sometimes hard to follow, please make the sentences clear and in the spirit of the English language.
  • The chemical formulas of the 6 scintillators often repeat in the text and the captions. It makes the text hard to follow. In my view it would be easier to follow some abbreviations. E.g. in Table 1 you could define Material 1, Material 2... and then later in the text refer to that. 

More detailed comments, with the corresponding line number (L):

  • Abstract: please point out the key results in the abstract
  • L16-17: it is not clear why the exposure times pose challenge, please be more specific
  • L21: "achievable" is not an appropriate word in this case
  • L22: "To suppress the effect...." please improve in the spirit of English language
  • L23: 2.5 s - what is the uncertainty of this value? Also add reference. 
  • L23: "previous" - should be removed
  • L25: studies on -> studies of
  • L25 and throughout the text: "have been performed up to now" OR "were performed in the past". The present perfect tense and the past tense are sometime mixed up. Please be consistent.
  • L30: "analyzed" -> "measured"?
  • L33: "phosphorous" - I find this word too colloquial, since those material do not contain phosphor. I would say "scintillating" or "luminiscent" material 
  • L34: "during" -> "for" ? 
  • L35-39: These sentences are too general and it is not clear what you want to say. Please be more specific.
  • The Results section: the whole section must be made more systematic, without repeating and without the discussion which belongs to Discussion section. The flow of the information must be generally improved. 
  • L42: "absolute" is incorrect. Throughout the paper you report about light yield in some a.u. Please explain what are these a.u. 
  • L53: 68% and 67% - must quote numbers with their uncertainties.
  • L53: "comparably" is not an appropriate word here
  • L55-L57: I would like to see the appropriate references quoted here.
  • L65: While an absorption -> While the absorption
  • L69: I believe the subtitle should read: "Initial and long term light yield during constant neutron  irradiation flux
  •  Figure2 - caption, 2.6 x 10^8 n cm-2 s-1 (green) - it this a mistake, should it be 2.5?
  • L76-82: it is hard to follow which material belongs to which group. 
  • L87: "which may be used" should be removed
  • L107: this has already been mentioned, no need to repeat. 
  • L134: variance -> variation
  • Figure 3 - caption: Relative light yield - please specify relative to what. Generally the caption is too large containing informaton that belongs in the text.
  • L146 Please explain why you think the noise may be the reason for the flattening of the curves. 
  • L147 Please elaborate why is this interesting
  • L159 "the highest noise" - that is not quite correct. I believe that the noise is the comparable in all data, just that in this case the relative contribution of noise is the highest. Also explain what is the source of the noise. 
  • Figure 4b. You plot the light yield for irradiation with different fluxes, but normalized to t=0. It would be interesting to see the "absolute" light yield with time for these different fluxes. This would give information on how sensitive is each screen to irradiated flux.
  • L168 "higher resolutions" - be more specific to which resolution you refer to: time resolution, energy resolution, spatial resolution...
  • L170, 171: you should explain the "units" of this light yield, since this is very ambiguous. 
  • L194-197:  please provide the supporting evidence for the claims, i.e. give references
  • L199: you mention that the number of traps does not change with temperature. One would expect that annealing is faster with the higher temperature. Could you comment on that.
  • L203: How does the noise change with temperature?
  • L212: you could express the proton flux with 1MeV neutron equivalent
  • L225: quantify "source fluxes"
  • Discussion section: missing comparison with other results from literature
  • Materials and Methods section: must be significantly extended to provide sufficient innformation for the reader to understand the experiment and how the results were obtained.

Round 2

Reviewer 3 Report

Dear authors, 

I am impressed by the work that has been invested in the significant improvements of the manuscript. 

Please find some additional minor comments:

L83. Please define all abbreviations, in this case FRM 

L83. Explain what is "white beam"

L87. Define L/D ratio

Figure 1. The name of the scintillators is overlayed on the histogram bars which reduces the clarity. Could you move the scintillator names above the lower of two bars for each scintillator, so it does not overlay with the histogram.

L129. "absorption" -> absorption rate or absorption percentage

L250. The sentence in somehow superfluous, that is clear that the properties are different if the materials are different.

L307 a mixture of -> a superposition of
